# Exploring Knowledge Domain and Emerging Trends in Climate Change and Environmental Audit: A Scientometric Review

**DOI:** 10.3390/ijerph19074142

**Published:** 2022-03-31

**Authors:** Guohua Qu, Yue Zhang, Kaichao Tan, Jiangtao Han, Weihua Qu

**Affiliations:** 1School of Management Science and Engineering, Shanxi University of Finance and Economics, Taiyuan 030006, China; hanjt813@163.com; 2School of Computer Science and Information Engineering, Harbin Normal University, Harbin 150025, China; tankaichao424516@163.com; 3School of Economics and Management, Shanxi University, Taiyuan 030006, China; 4Institute of Management and Decision, Shanxi University, Taiyuan 030006, China

**Keywords:** climate change, environmental audit, knowledge map, public health

## Abstract

Environmental audit is inevitably linked to climate change, one immediate target of the auditors is likely to be climate control, and the warming of the Earth and the consequent climatic changes affect us all. What is the link between environmental audit and climate change? What ties together some of these themes between environmental audit and climate change? The interaction between climate change and environmental audit has been one of the most challenging. In this paper, a scientometric analysis of 84 academic publications between 2013 and 2021 related to climate change and environmental audit is presented to characterize the knowledge domain by using the CiteSpace visualization software. First, we present the number of publications, the number of citations, research categories, and journals published through Web of Science database. Secondly, we analyze countries, authors, and journals with outstanding contributions through network analysis. Finally, we use keyword analysis and apply three types of knowledge mapping to our research, cluster view, timeline view, and time zone view, revealing the focus and future directions. We identify the most important topic in the field of climate change and environment audit as represented on the basis of existing literature data which include the Carbon Emissions, Social Capital, Energy Audit, Corporate Governance, Diffusion of Innovation Environmental Management System, and Audit Committee. The results show that climate change and environmental audit publications grew slowly, but the research are widely cited by scholars. Published journals are relatively scattered, but the cited journals are the world’s top journals, and most research countries are developed countries. The most productive authors and institutions in this subject area are in UK, Australia, USA, Spain, and Netherlands. There are no leading figures, but the content of their research can be divided into six clusters. Future research content involving city, policy, dynamics, information, biodiversity, conservation and clustering social capital, diffusion of innovation environmental management, and audit committee are the directions for future research. It is worth noting that cities, policies, and adaptability are closely linked to public health.

## 1. Introduction

Climate change is an issue that determines the characteristics of our time, and we are at a decisive moment. Climate change is mainly manifested in global warming, acid rain, and ozone depletion, among which global warming is the most urgent problem for human beings. Climate issues are related to the future of mankind. Changes in weather patterns will threaten grain yield, rising sea levels will cause catastrophic floods, and extreme climates such as blizzards, heavy rains, droughts, hail, and typhoons will occur frequently across the Earth [1]. In addition, global warming increases the likelihood of epidemics and other public health threats. There is no denying that the coronavirus disease (COVID-19) pandemic is related to the climate crisis [2]. However, because of anthropogenic global warming and pollution, we are in a state of climatic emergency, facing huge ecological and humanitarian catastrophe, far exceeding the current COVID-19 epidemics, and these effects are faster than expected and more dramatic. In 2013, the Intergovernmental Panel on Climate Change issued its fifth assessment report, which stated that from 1880 to 2012, the global average temperature rose by 0.85 °C, and human activities were the main reason for this. In a report released by the IPCC in October 2018, it was stated that global warming must be limited to 1.5 °C [3,4].

The climate issue has become very urgent, and we must take emergency measures to prevent climate change and monitor the climate in real time. Environmental audit is an innovative technical management, which means that the organization can eliminate the damage that its activities may cause to the environment at the beginning of its activities, instead of eliminating the pollution after the activities cause environmental pollution [5]. Since the 1960s, environmental audit has been widely promoted in various countries around the world [6]. Countries have also promulgated corresponding policies to promote the development of environmental audit, and enterprises have their own environmental audit plans. Environmental audit plays an important role in the management, supervision, and evaluation of the environment.

There is an inevitable internal relationship between climate change and environmental audit. Serving to improve the climate is the trend and inevitable choice of the development of environmental audit. Environmental audits are an unavoidable requirement for reviewing and evaluating the exercise of power, the allocation of public resources, the improvement of the living environment, and the expansion of environmental carrying capacity. With the quantity of research on this background, various angles have been presented in these analyses, such as through environmental audit to reduce the environmental impact of dairy factories to improve the climate [7]; environmental policy regulation and corporate compliance in game model [8]; auditing of the environmental impact assessments for mining projects [9], the relationship between residential areas, environmental carrying capacity, and continuous spatial management [10]; urbanization perceptions and reactions to the impacts of carrying capacity of natural resources [11]; and quantitative approaches to evaluating the effects of climate change on urban expanse [12]. Despite a wide range of publications related to climate change and environmental audit, most of which are based on research by different scholars from their own professional background, this subjectivity and individual research has led to a lack of quantitative analysis in this field. This will lead to blind and inefficient audits instead of more efficient and targeted audits on the most climate-affected aspects.

In fact, there are no known articles that address the overall structure of its knowledge domain. Therefore, it is very necessary to give a general overview of the situation of publications on climate change and environmental audit to explore the intellectual structure of this field and its possible development trend by focusing on the scope, core author, core journal, and high frequency keywords.

The domain of mapping knowledge is to present the knowledge in the form of charting, mining, analysis, and display. The scientific knowledge graph is not only a visual knowledge graph but also a serialized knowledge map, which shows the relationship between network, structure, intersection, and evolution among knowledge groups. Compared with other visualization tools, CiteSpace, a relatively widely used information visualization tool, has the advantages of simple operation, diverse maps, large amount of information, clarity, and high interpretability of visualizations [13]. Additionally, it provides a variety of analysis methods, such as collaboration network analysis, co-citation analysis, and keyword co-occurrence analysis. Information visualization is the use of graphics and image technology to help people intuitively and clearly understand and analyze data. Therefore, this study uses the CiteSpace software to conduct a comprehensive and quantitative visual analysis of the climate change and environmental audit literature.

Despite the popularity of CiteSpace, to our best knowledge, no attempt has been made to use CiteSpace to analyze the expanding climate change and environmental audit literature. In order to provide a systemic and objective overview of research, the scientometrics and information visualization methods will be used to analyze the literature on climate change and environmental audit between 2013 and 2021 based on the Web of Science and CiteSpace. Specifically, the research is guided by three key goals: (1) to understand the basic status and development process of climate change and environmental audit; (2) to illustrate the most influential knowledge groups and their key contents in the past nine years from the perspectives of countries, authors, and journals; and (3) to gain research hotspots from different research perspectives (clustering, high frequency keywords, keyword time zone) and implement more targeted environmental audit in the future.

The rest of the paper is organized as follows: In Section 2, we describe the process of data resources collection, the analysis methods of this article, and the analysis tool advantage. In Section 3, we make a detailed explanation of the results produced with regard to the research focus and research font cluster according to CiteSpace. In Section 4, we focus the significant results on different varieties of evolution network of climate change which is affected by environmental audit.In Section 5, we propose the main further development of climate change and environmental audit. In this way, it is possible to let researchers enrich and expand the related scientific research on climate change and environmental audit and provide long-term effective scientific guidance and efforts for making a contribution to tackling climate change by environment audit and environment management to set up and take the feasible measure.

## 2. Materials and Methodology

### 2.1. Data Collection

The reference used for analysis in this study was downloaded from the Web of Science (WOS) Core Collection database. WOS is a database of the Institute for Scientific Information (ISI), which contains more than 13,000 authoritative and high-impact academic journals worldwide and comprehensively covers the most important and influential libraries of integrated academic information resources in the world. In addition, the WOS core collection database content covers areas such as social sciences, natural sciences, engineering, technology, management, and biology. Although Scopus is broader than WOS in terms of coverage, Scopus has high data overlap. Archambault et al. used WOS and Scopus in their analysis, and they found than there were no significant differences in their results [14]. Therefore, it is reasonable and effective to use the WOS database as the data source. In order to make the knowledge maps clearer and easier to understand, we only chose English articles. Only 84 papers related to climate change and environmental audit were published between 2013 and 2021 by a topic search using the terms “climate change” and “environmental audit* or ecological audit* or green audit” in titles, abstracts, or keywords; the wildcard character “*” indicates that any words beginning with the preceding letters should be taken into account. The keywords retrieval refers to the expressions of international authoritative journals related to climate change and environmental audit. Export all its records (such as author, title, abstract, keywords, cited references, etc.) to a text-based format and convert it into a data format that CiteSpace can process. Although the environmental audit appeared early, the technology is still immature and there are some difficulties in the implementation process. There has been an increase in the number of articles published in recent years analyzing the impact of environmental auditing on climate change. However, there is currently no quantitative or visual research on climate change and environmental auditing, implying that the audits still lack a comprehensive understanding. The reason for choosing this period (2013–2021) is that the attention has changed since 2013, and it has become one of the important research fields for researchers. Furthermore, certain journals in WoS had older publications that were not available. Therefore, it is feasible to study the literature of the past nine years. We created a statistical summary of all publication keywords and show their frequency and centrality in Table 1. Data update time is 25 October 2021.

### 2.2. CiteSpace

CiteSpace is a scientific bibliometric analysis tool built with the Java programming language. This innovatively combines citation analysis and co-citation analysis methods to develop a theoretical model that maps from “knowledge base” to “knowledge frontier”. Scientometrics method is an objective quantitative research method that uses statistics and other mathematical methods to quantitatively analyze the input and output of scientific activities and find out the rules. Scientometric mapping is a visual technique in informatics that quantitatively displays structural and dynamic aspects of scientific research. CiteSpace is easy to use, provides a lot of information, and is simple to understand when compared to other visualization software. It offers a number of analysis methods, including collaboration network analysis, co-citation analysis, and keyword co-occurrence analysis, as well as three visualization methods such as cluster, timeline, and time zone, avoiding the subjectivity generated by qualitative analysis. Through the drawn knowledge map, the evolutionary process of a knowledge field can be collectively displayed on a citation network map, and key information nodes on the map are automatically represented. For instance, after the visualization of input data through CiteSpace, we can illustrate the intellectual structure and internal relationship of the environmental auditing domain. It offers greater clarity and network translations than other visualization software, bringing together the dynamics and architecture of a specific field in a visual diagram. It has been used in various fields since software development, such as climate change and tourism research [15], Western economic geography research [16], education administration [17,18], online health [19], and so on.

### 2.3. Collaboration Analysis

Collaboration analysis can reveal connections on both an intellectual and a social level. Groups working in the same intellectual area and socially cohesive groups can be identified [20]. The co-author clusters appeared to be meaningful in terms of identifying research groups, the relationships within these groups, and the partnerships between these groups and changes over time [21]. The analysis of country cooperation networks shows the distribution of areas of cooperation and the intensity of cooperation between individual countries, indicating which countries frequently appear together in the same paper. The color of the lines indicates the year cited, with warm colors indicating the most recent year cited. The size of the nodes and the font size in the resulting map represent the number or frequency of references. For example, a record shows that a 2018 article, entitled “How are cities planning to respond to climate change? Assessment of local climate plans from 885 cities in the EU-28” [22], has authors from the two countries (University of Newcastle, Sch Engn, Tyndall Ctr Climate Change Res, Newcastle Upon Tyne NE1 7RU, Tyne & Wear, UK; University of Reims, 57 Rue Pierre Taittinger, F-51571 Reims, France). In this case, the United Kingdom and France would be connected by lines in the country collaboration network. The collaboration analysis in this study focuses on identifying the research community and major countries in the climate change and environmental audit field.

### 2.4. Co-Citation Analysis

The frequency with which two previous pieces of literature are cited together by later literature is referred to as co-citation [23]. Co-citation analysis is often used to reveal the relationship and structure of authors, articles, and journals in academic fields. This paper analyzes author co-citation networks and journal co-citation networks based on environmental auditing and climate change area to explore the underlying clusters that have high article co-citation counts associated with them as well as the most cited authors or journals to find the powerful points in the knowledge structure. We chose the top 50 most-cited items in each time slice to study their intellectual structure and the dynamics of co-citation clusters for the co-citation analysis example indicated above, and we established a time period from 2013 to 2021. CiteSpace can also compute the values of modularity Q and mean silhouette based on the overall structural properties of the network. The value of modularity Q describes the divisional quality of clusters, and it ranges from 0 to 1. If the value is greater than 0.3, the network structure is considered significant. In order to make the knowledge network more concise and clearer, we use “Minimum Spanning Tree” to prune (set as base setting). In “cited author network”, we select cluster labels by index terms from their own citers and log-likelihood ratio (LLR) weighting algorithm. LLR is an algorithm to calculate and determine each type of labels, which presents a core concept of each cluster with professional words [24].

### 2.5. Keyword Analysis

Keywords are concise summaries of articles and precise descriptions of their core contents, which can reflect hot topics and emerging trends in research. The evolution of research topics has been paid attention to by many scholars, and it provides researchers with ideas about the existing achievements and future directions of a specific field. CiteSpace not only provides a deeper understanding of the research object, but also identifies future research areas. The analysis based on CiteSpace has two main parts, firstly, extracting keywords to calculate the frequency of occurrence, secondly, obtaining keyword time zone graph analysis to monitor more intuitively the evolution trend of the 9-year time span from 2013 to 2021. The analysis based on CiteSpace has two main parts, the first is to extract keywords to calculate the frequency of occurrence, and the second is to obtain keyword time zone graph analysis to monitor more visually the evolution trend of the 9-year time span from 2013 to 2021 [21,24]. It is used to look forward to future research directions.

## 3. Results

In this section, we present the results of this paper in details in terms of three parts. In Section 3.1, we count the current status of climate change and environmental audit research. In Section 3.2 and Section 3.3, we introduce the main contributing countries and use the CiteSpace software for co-citation analysis, cluster analysis, and keyword analysis. Figure 1 shows the schematic diagram of the study.

### 3.1. Research Outputs and Their Categories

#### 3.1.1. Research Output

The number of papers published related to climate change and environmental audit during the 9-year period between 2013 and 2021 is shown in Figure 2. Since the data retrieval is from 25 October 2021. From Figure 2, we can see that since 2013, the number of publications on climate change and environmental audit has been on the rise, but the base of the number of publications is relatively small. From 2013 to 2021, the rate of growth was relatively slow and the number of publications per year remained below 20. It shows that there are more studies on climate change or environmental audit, but the analysis of the combination of the two is still relatively small and few articles systematically study the role of environmental auditing in climate change. The number of publications is on the rise, indicating that people are increasingly aware of the importance of environmental audit to climate change.

#### 3.1.2. Research Categories

Publications on climate change and environmental audit covered various subject categories in the Web of Science. In order to understand the distributions of subject categories and the dominant categories, we present the data in a more visual way on area charts so that the disciplines in the development of the research area can be clearly identified.

The top 10 research categories and the proportion of publications in this field are shown in Figure 3. From Figure 3, we can know that the most researched category is environmental sciences ecology, which shows that the research focus on climate change and environmental audit is ecology. The ecological environment is an organic ecosystem centered on human beings and linked by materials, energy, and information. Once the ecological environment is destroyed, the homes on which we humans depend will be destroyed. The biggest impact of climate change is the ecosystem, so the focus of environmental audit is also the focus of ecological environment research. In addition, business economics is also an important category. Environmental audit is an audit by a government or business of activities performed to avoid pollution to the environment. As a unit operating for profit, an enterprise must consider the economic feasibility when conducting environmental audit. The government must consider maximizing social benefits. Therefore, in response to these categories, many scholars have carried out research to find the optimal environmental audit solution from a different perspective. The energy issue is also a key concern of every country. Human survival and development depend on the supply of energy. We must achieve accurate and efficient energy audits. In particular, the audit of fossil-based energy has a great impact on the climate. We must focus on control and prevention.

#### 3.1.3. Research Journal

Figure 4 shows the top 10 journals and publications in this field. As can be seen from Figure 4, the journal with the largest number of articles is “Science of the Total Environment” with only four articles, accounting for 5% of the total. The journals with the second highest number of articles are “Business Strategy and the Environment”, “Journal of Cleaner Production”, “Renewable Sustainable Energy Reviews”, “Resources Conservation and Recycling”, and “Water”, all of which have three articles and only 4% of the total articles. We can see that the publications related to climate change and environmental audit are not fixed in a specific journal, and their distribution is relatively scattered. Combining these journals, it can be seen that the main research directions of the journals on climate change and environmental audit are environmental science, energy and fuel, ecology, and water resources. This corresponds to the above research direction, and these are also the focus of climate change and environmental audit research. In addition, the journals with two articles is “Energy and Buildings”.

### 3.2. The Collaboration Network and Co-Citation Network of Climate Change and Environmental Audit Research

#### 3.2.1. Country Collaboration Network

The country collaboration network represents the cooperation relationship between different countries and the countries with the most papers on climate change and environmental audit. The network consisted of only 41 nodes and 48 links, and the network density is 0.058, as show in Figure 5.

From Figure 5, we can see that the largest number of nodes is in UK, and its centrality is also the highest. It shows that the United Kingdom has contributed the most to climate change and environmental audit, and is also the country that attaches the greatest importance to the climate. Europe and its cities are sources of substantial greenhouse gas emissions. With the development of the times, these countries have now become the main participating countries in environmental protection. In recent years, European cities have played a large role in improving climate levels, formulating global climate plans, and making policy decisions [22]. Followed by Australia and the United States. We can find that studies on climate change and environmental audit have been predominantly conducted in developed countries, and studies in developing countries are lacking. Developed countries have passed the period of heavy industry that pollutes the environment. Production technology is relatively advanced, and the pollution caused to the environment is relatively small compared to developing countries. Developed countries discovered more significant technologies and have stronger innovative capabilities in this specific domain. What these countries are doing now is to research new technologies to improve the climate and protect the environment. For some developing countries, they are still in the period of industrial development, which will inevitably cause pollution to the environment. What they need to do at this time is to reduce or eliminate the pollution to the environment during the development process. Developing countries should pay more attention to the quality of theses in order to increase their international academic influence and strengthen cooperation with other countries. Since 2007, China has become the largest carbon emitting country in the world [25]. Therefore, as China is the largest developing country, the study of climate change and environmental auditing in China is essential. After nearly 30 years of development, China’s environmental audit has evolved from single to diversified. It has successively carried out environmental audit in environmental protection funds, ecological construction, water pollution prevention in key river basins, environmental impact of construction projects, and air pollution prevention. In addition, there are also countries such as the Netherlands, Spain, Sweden, and India who have contributed significantly to climate change and environmental audit. The country details are shown in Table 2.

#### 3.2.2. Author Co-Citation Network

The author co-citation network represents the high contributing authors in climate change and environmental audit. The network consisted of 41 nodes and 39 links, and the network density is 0.0476, as shown in Figure 6.

From Figure 6, we can see that the most cited authors are United N, Eurostat, Schaltegger S., Gossling S., and Adams C. United N has mainly published two reports on the environmental effects of ozone depletion and its interactions with climate change: Progress reports, 2015 [26] and 2016 [27], cited by many scholars. These two reports mainly explain the effects of ozone depletion and the consequences of climate change interactions with respect to human health, animals, plants, biogeochemistry, air quality, and materials. Gossling S. [28] mainly studied the impact of carbon emissions on climate in the German transport sector. In the transport sector, global climate policy is of vital importance. Schaltegger S. [29] focuses on the environmental impact of companies using sustainable management tools. The findings show that the implementation of sustainability management tools does reduce environmental impacts per unit of revenue. Adams C. [30] mainly studies the impact of the adoption of environmental management accounting (EMA) by corporate water supply organizations from an institutional theory perspective. Eurostat is an agency that counts European data and one of the main sources of statistical climate data. Zobel and Zografak have published fewer articles, but have a higher centrality, indicating that they are authors with a strong influence in the field of climate change and environmental audit. Table 3 shows the details of the top 10 highly cited authors.

The main clusters in the climate change and environmental audit research area are shown in Table 4. We can find that the network is divided into six co-citation clusters. A timeline view of the network of each cluster is arranged on a horizontal timeline. The direction of time moves to the right. This timeline map showed some of the basic movements of the research area. According to the timeline map observed in Figure 7, we can see that the year in which the keyword clustering began to appear was when the clustering results began to increase and the trending of the development of the whole cluster can be seen. The clustering modularity value is 0.7068 > 0.3, indicating that the clustering is significant. The mean silhouette is 0.7291 > 0.7, which indicates that the clustering is efficient and convincing. The largest cluster is cluster #0, and the smallest cluster is #5.

The largest cluster (#0) is carbon emissions, with a time span from 2013 to 2019. Carbon emissions have an important impact on climate change, and are one of the main causes of global warming and glacial melting. CO_2_ is one of the most important components of greenhouse gases. Between 1959 and 2008, the average annual CO_2_ emissions were about 43% of total emissions. In order to control global warming, CO_2_ concentrations must be stabilized [31]. To improving the climate, we must strictly monitor carbon emissions. The environmental audit must also focus on auditing carbon emissions. This requires enterprises to use low-carbon technologies to improve production processes. Carbon capture and storage (CCS) technology is considered to be an effective way to reduce the emissions of CO_2_, it can capture and sequestrate the carbon dioxide [32]. This technology can reduce CO_2_ emissions from fossil energy by more than 65%. The environmental system is an open and complex giant system. To solve the problem of “carbon neutrality and carbon peaking”, we need to apply system thinking, consider and plan from a global perspective, and build an ecological civilization. For developed countries, the industrialization process is a major factor in carbon emissions, and promoting emission reductions is conducive to their export technologies and will improve their international competitiveness. For developing countries, reducing carbon emissions will limit their economic growth space. For enterprises, the market value of the most carbon intensive firms decreased by 7% to 10% compared with other firms in the market assesses [33]. This requires countries and businesses to find a balance between economic and environmental development. For us ordinary people, we need to achieve a low-carbon life, save energy, and travel with low carbon.

The second cluster (#1) is social capital, with a time span from 2018 to 2020. This refers to the total social capital spent by the government, enterprises, and the public as the main actors in society to improve the environment. As a regulatory agency, the government pursues social benefits. Enterprises are the main source of pollution to the environment, and their pursuit is to maximize profits. The public is a major participant in society. Coordinating the relationship between the three can greatly improve the effect of environmental audit. Many scholars are committed to conducting research from the perspective of business economy, so that more companies can participate in environmental audit. However, the government’s guidance to consumer consumption and punishment to polluting enterprises play an important role in whether enterprises join third party international environmental audit [34]. Social capital has a lot to do with corporate governance. The government must also reduce the social impact on the environment by developing sound policies. Against this background, it is very necessary to find a better way to improve the environment with the least social capital.

The third cluster (#2) is energy audit, with a time span from 2016 to 2021. Energy audits are mainly audits of traditional non-renewable energy and environmentally harmful energy. Fossil-based fuel energy production will lead to increased CO_2_ and cause serious climate change issues [35]. This requires countries to actively explore and apply new types of clean energy and reduce the use of environmentally harmful energy. In the activities of a transnational programme of the European territorial cooperation (MED Programme) financed by the European Union, these countries are actively exploring offshore wind power generation and other offshore renewable energy technologies, such as tidal barrage and wave energy converters, etc. [36]. China is the country with the most carbon emissions and is the largest coal country. Its coal-to-electricity policy has a major impact on the economy, energy, and environment. Nowadays, the heating method of residents is gradually changing from coal burning to electric heating, which can help reduce carbon dioxide emissions and improve the environment [37]. Electric energy is a new type of clean energy, and its application is very mature. Many cities have started to promote the use of electric cars, which will greatly improve our environment. However, in the process of promotion, there are still many problems waiting for us to solve. In addition, new energies such as solar energy, geothermal energy, wind energy, ocean energy, biomass energy, and nuclear fusion energy are yet to be developed and used by us.

The 4th cluster (#3) is corporate governance, with a time span from 2014 to 2021. Corporate governance mainly analyzes the corporate’s behavior from economic, management, and philosophy [38]. From the perspective of economic feasibility, the enterprise aims to maximize profits. At the same time, it will inevitably cause damage to the environment. If the company considers environmental issues, it will inevitably have an impact on profits. The management mode is a management plan put forward by the enterprise’s senior leaders on practice and needs. Generally, whether an enterprise is more environmentally friendly has a lot to do with the management mode of senior leaders. The enterprise audit should be conducted from four aspects, mainly the enterprise’s compliance with relevant regulatory requirements, the enterprise’s compliance with industry standards, the enterprise’s management of daily environmental matters, and the enterprise’s actions to correct identified deficiencies [39]. This requires corporate managers to manage the company with a higher standard within the scope of the law, and often conducts introspection of corporate behavior. The philosophical point of view is from the moral point of view, it is a new method. There is no standard for judging this moral perspective, which is mainly reflected in the corporate behavior and its social responsibility. The economic method is the main method in academic history, and a philosophical perspective may be the focus of future research.

The 5th cluster (#4) is diffusion of innovation environmental management system, with a time span from 2013 to 2019. An innovative environmental management system requires an intelligent environmental management system to mitigate environmental impact while ensuring its commercial viability [8]. There is little research in this area because of the large uncertainty of complex social network relationships. Currently, big data and artificial intelligence are hot topics. It can use powerful data analysis capabilities to drive the relationships from complex abstract networks. This is a good tool to apply to complex environmental systems. Therefore, scholars need to conduct in-depth research in this area, combine new renewable energy sources, discover new technologies, and explore a suitable environmental management framework.

The 6th cluster (#5) is audit committee, with a time span from 2014 to 2020. It is necessary to establish a special environmental audit department, train more professional environmental experts, and formulate more effective and authoritative auditing standards. A more professional approach to environmental auditing can maximize climate improvement. The environmental audit committee can supervise and inspect the company’s environmental audit work and strengthen the authenticity and reliability of audit information. Provide more accurate environmental information for government and business decisionmaking. Currently, many accountants are competing for environmental audit work. It shows that there are many similarities between environmental audit work and accounting work, but in order to become a professional environmental auditor and perform more accurate environmental audit work, more professional training is required.

#### 3.2.3. Journal Co-Citation Network

The journal co-citation network represents the set of journals that have served the climate change and environmental audit research community. By analyzing the citation frequency of core journals, the quality level of publications from a given journal can be effectively revealed. The network of co-citation journals consisted of 126 nodes and 169 links and the network density is 0.0215, as shown in Figure 8.

From Figure 8, we can see that the most-cited journals are “Journal of Cleaner Production” and “Science”. The journal distribution of the cited literature represents the subject content cited in climate change and environmental audit research. Among them, “Journal of Cleaner Production” is focusing on environmental science and ecology. “Science” is one of the top academic journals publishing original research across a wide range of scientific fields, and there is also “Nature”. These two most central journals illustrate the authority of climate change and environmental audit research. This shows that scholars are paying more and more attention to the role of environmental audit in climate. Climate and environmental issues are also key international research areas. In addition, in “Energy Policy” the main research direction is energy and fuel, which also corresponds to the research direction and research journal above. The top 10 co-citation journals are listed in Table 5.

### 3.3. The Keywords Network of Climate Change and Environmental Audit Research

#### 3.3.1. Keyword Analysis

Keywords are a high-level summary of an article and the keywords with high frequency are the research focus in this field. In order to gain a more comprehensive understanding of climate change and environmental audit, the co-occurrence network of keywords is shown in Figure 9. The network of co-occurrence consisted of 31 nodes and 25 links and the network density is 0.0538. The top 12 keywords are shown in Table 6.

According to the keyword co-occurrence map, these keywords can be divided into three topics for further discussion. The first topic is centered on climate change. From Figure 9, we can see that the keyword node of climate change is the largest and has a close connection with other nodes. Related to this keyword are city, impact, policy, adaptation, etc. These are the key factors affecting climate change and countries that have made a significant contribution in climate change. As highly centrality words, city and policy have attracted more and more attention in recent years. For example, how policymakers can promote the interaction between corporate environmental compliance and enforcement [40], and how should cities plan to respond to climate change [22]. Because cities are at the interface of local action and national and international level climate change adaptation and mitigation commitments, cities play a key role in developing and implementing climate change programs [41].They study the climate from the aspects of city development, policy formulation, and economic growth, and are committed to conducting audits from different perspectives in order to achieve the purpose of improving climate. Australia and Europe are countries that study climate change the most, and correspond to the country collaboration network above.

The second topic is based on management and performance. Centrality of management is 0, indicating that they are not closely related to other keywords. From a management perspective, enterprises and governments can improve management methods and innovation management models to reduce economic expenditure and reduce total social capital in a better way. For example, improving sustainable production programs for climate change in the industry to improve resource efficiency [42], increasing the credibility of sustainability report through audit committees [43], promoting environmental performance and social responsibility through disclosure of greenhouse gas emissions requirements [44], and developing a new theoretical model to improve energy efficiency and reduce carbon emissions [45]. They studied management methods and built management models to improve the overall performance of enterprises and society, and eventually achieve the goal of climate improvement.

The third topic is based on information and biodiversity. They have fewer nodes, but have a high centrality, indicating that they are closely related to other keywords. Among them are efficiency, dynamics, and conservation. Information is mainly reflected in information asymmetry and information differences. There is a gigaton gap between China’s national carbon dioxide inventory and the summation of provincial inventory data between 1997 and 2010 [46]. Asymmetry information makes investors lack essential environmental information, which leads to incomplete information for investment decisions and misallocation of financial markets funds in financial markets. By solving the problem of information asymmetry, environmental pollution can be resolved [47]. Therefore, the accuracy of carbon emissions detection information and solving the problem of information asymmetry have become critical. In addition, the impact of black carbon from incomplete combustion of carbon on the environment is also very bad. We also need to improve the efficiency of carbon combustion to make it fully burn. In terms of improving efficiency, the use of new technologies can greatly help improve energy efficiency, such as promoting energy conservation behaviors through thermal imaging [48]. In terms of industrial carbon, it is necessary to improve industrial energy efficiency measures to reduce industrial energy consumption [49]. At the same time, in terms of biodiversity, the extinction of species will lead to a decrease in biodiversity and a negative impact on the ecological environment [50].

#### 3.3.2. Keyword Time Zone Map Analysis

In order to explore research directions, grasp the research focus, and display the research content of climate change and environmental audit from multiple perspectives and fields, a time zone view of keywords is illustrated in Figure 10 and this visualization view arranges keywords in correspondence to the time of their publication. Figure 10 clearly shows the evolution of the keywords path from 2013 to 2021. Since the data update time is 25 October 2021, no critical node appears at this time. From the map, we can see the hot topics in the fields of climate change and environmental audit at each time. In 2013, the node of climate change was the largest, and at the same time, it was closely connected with the keywords of other periods.

Climate change first appeared with the organization and Australia, illustrating that Australia as a developed country contributed early on to climate change and environmental audit. Then, in 2014, performance, management, and information, indicating that at this time people have taken enterprise management and performance as the important research content of environmental audit, and are exploring efficient and economical innovative management models to improve the climate. In 2015 and 2016, two keywords appeared: forest and influence, indicating that there were no representative research results at this stage. Many researchers have studied large-scale forest fire cases, and studied the impact of forest fire carbon emissions on climate change. It can be seen that the growth of research topics occurred mainly in 2017, and major keywords such as city and policy appeared. The formulation of government policies and city development planning play a vital role in the environmental audit of enterprises. At the same time, carbon emissions have also become a research hotspot. The research focuses on the accuracy of corporate carbon emissions audits. In 2018, keywords such as corporate social responsibility and strategy appeared. We can see that corporate social responsibility and corporate strategic planning have a great impact on the climate. Enterprises should strengthen their social responsibility, join the ranks of environmental audit from a strategic perspective, and pursue longer-term development. It is necessary to strengthen the awareness of managers and the implementation of innovative tools, make them voluntarily adopt environmental governance mechanisms, and set higher emission reduction targets for companies. Scholars should try to study corporate behavior from a new perspective, such as philosophy.

In 2019, more intensive keywords appeared, among which biodiversity, conservation, and dynamics are the most central nodes. Information is mainly reflected in information asymmetry and information differences. At this time, people have focused on the impact of biodiversity on the climate, mainly manifested as the impact of forest ecosystems on the environment after forest fires, and the importance of protecting biodiversity. The ecosystem is very large and complex, but once it is damaged, it cannot be recovered, which brings great disaster to humankind. The ecosystem is an organic whole composed of biology and environment. Its complexity makes research very difficult, and it also requires us to spend more time and energy on it. Water resources are also a major research focus, focusing on salinity and groundwater. At this time, European countries have a stable cooperative relationship and play an important role in climate change and environmental audit.

## 4. Discussion

Knowledge mapping can provide us with valuable information about the status and trends of a certain field. Literature mining and analysis can not only quantitatively analyze and sort out the literature in this field, but is also useful for researchers to find new research directions. In a research area, often unexpected information is more valuable. In the past, most of the research focused on specific topics in climate change or environmental audit research, and did not analyze the complete research field [51,52,53]. In recent years, there have been some studies that analyze the impact of environmental audits on climate change, but other researchers have not conducted similar quantitative analysis of this area [54,55]. It is the first study to use the scientometrics and information visualization methods to conduct a knowledge structure analysis in this area. This paper focused on the research field of environmental audit for climate change, addressed the research directions, the developing status, and future research direction. Therefore, more specific analysis of the intellectual dynamics of different timeline stages, and the high-frequency keywords and key clustering to indicate the increasing research directions under the background of climate change is needed.

Through the basic summary of climate change and environmental audit research, from the perspective of research output, there are relatively few publications on climate change and environmental audit. However, the research questions have great practicality and are widely cited by scholars. At the same time, various scholars cited each other, reflecting the characteristics of interdisciplinary research in this field. Simultaneously, it has also promoted the development of environmental audit. From the perspective of research direction, the research direction suggested the high priority of environmental sciences ecology and business economics in this field. In addition, energy fuels are also areas that deserve special attention. Moreover, climate change and environmental audit research become more interdisciplinary over time. At the same time, these areas also provide new possibilities and greater space for the development of environmental audit. From the perspective of research journals, the number of publications in each journal is very small, even if only a maximum of four. The small number of articles in various categories shows that although environmental audits were proposed early on, little research has been carried out on their actual benefits. The small number of theme and journal publications shows that there is no centralized research trend in this area, and the development of climate change and environmental audit is in its infancy. Scholars should focus on this aspect, explore better environmental audit methods, and pay attention to the result of the impact on the climate after the implementation of environmental audit, and control the climate more efficiently, rather than blindly auditing. Climate change is a concrete reflection of the benefits of environmental auditing, so it is important to study climate change and environmental audit. 

Through the country collaboration, there are few countries that combine climate change and environmental audit, indicating that most countries have not paid much attention to the impact of environmental audit on climate change. Among the countries that publish more documents, most are developed countries and studies in developing countries are lacking. Through the research in this article, it is hoped that all countries can strengthen the importance of environmental audit on the impact of climate change, and at the same time strengthen cooperation between various countries. Climate change and environmental audit are the common mission of all countries in the world. From the author co-citation network, these highly cited scholars explained the environmental audit work and studied the degree of its impact on climate change from different angles. In general, the highest number of citations was six and there were no scholars with extremely high citations. This shows that there are no leading figures in the fields of climate change and environmental audit. At this time, scholars should seize the opportunity and strive for major discoveries in this area. However, from another point, although the authors have not been cited frequently, they all have stable cooperative relations, forming a concentrated scientific research cooperation circle. It shows that the cooperation between authors in the fields of climate change and environmental audit is strengthened, which is conducive to the further exchange and sharing of knowledge. The clustering map mainly shows the main research content of these cited authors. Specifically, the hot spots in the field over the last 9 years are carbon emissions, social capital, energy audit, corporate governance, diffusion of innovation environmental management system, and audit committee. In addition, in the cluster analysis, we also combined some current research progress literature to further explain clustering. They may be emerging forces in this field and may become the mainstay in the future. From the journal co-citation network, the cited journals of climate change and environmental audit in the past few years are mainly the world’s top journals, and in recent years, they have diverged outward. This trend illustrates the diversity and fragmentation of climate change and environmental audit research and no highly concentrated research content.

Combining keyword co-occurrence and keyword time zone view analysis, the highly central keywords city, policy, dynamics, biodiversity, and conservation are the key research contents of climate change and environmental audit research in recent years. Among them, dynamics, information, biodiversity, and conservation are new keywords, and the centrality is around 0.4. Information is mainly reflected in information asymmetry and accurate measurement of carbon emission information. Future audit work in this area is mainly reflected in the study of information accuracy, timeliness, and transparency, so that it is more conducive to decisionmakers to master more environment information to make more informed decisions. Biodiversity and conservation are mainly reflected in the protection of the ecological environment and biodiversity to increase the dynamic adaptability of the environment. Although these aspects are emerging keywords in the field of climate change and environmental audit, in fact, protecting the ecological environment and biodiversity is the work we have been doing all the time, and it will be better in the future, combining with environmental audit to improve the climate. Global warming, reduced biodiversity, and water and air pollution have become health problems. Dynamics are faced with complex and changing environmental problems, and audit work must also be carried out flexibly to adapt to changes in the dynamic environment, and the environmental problems to be faced in the future will be more complicated, so environmental audits must continue to innovate and constantly improve, so that the future environmental audits can be carried out more scientifically and flexibly. Although city and policy are the keywords that appeared in 2017, environmental auditing has been constantly developing, and policies indicate the direction of its development and provide strategic planning goals. The development of cities is always the focus of environmental audits and the main factors affecting climate change. Climate mitigation and adaptation measures for cities are constantly widening, advocating the combination of biophysical characteristics and sociopolitical factors [56].

In addition, cities, policies, and adaptability are closely linked to public health. At present, more and more people are concerned about public health, which is related to the public health of the people of a country or a region. Among them are the prevention, monitoring, and treatment of major diseases, especially infectious diseases. In particular, the recent emergence of COVID-19 has raised higher requirements and challenges for the development of public health in various countries and cities. During the epidemic of COVID-19, the policies of different countries and the development of public health played a vital role in the prevention and control of the epidemic. The COVID-19 outbreak infects a large number of people, threatens the core of society, and has a devastating impact on society and the economy. For example, unemployment, vacation aggravate, and the decline in workers’ income and consumer consumption. The implementation of government policies and systems is essential to prevent such emergencies. Therefore, the establishment of a perfect public health prevention and control system is essential to the development of a country. In terms of environmental audit, from the perspective of the city, it is mainly reflected in the development of urbanization, the improvement of infrastructure, urban environmental and greening, and urban population changes, including population growth and population aging. In future urban planning, the space patterns should be optimized, more attention should be paid to the city’s adaptability and ability to respond to major disasters, and the city’s impact on the climate should be addressed. From the policy perspective, it is mainly through the implementation of national and local energy conservation and emission reduction policies to reduce the concentration of greenhouse gases in the atmosphere, reduce the burning of fossil fuels, and establish an adaptive mechanism to deal with those inevitable health consequences. Adaptability is to anticipate and prepare for the effects of climate change, thereby reducing the associated health burden. The COVID-19 crisis also led to a reduction in greenhouse gas emissions this year. On the contrary, after the crisis, once the economy is restored, emissions are likely to rebound, so the government must formulate appropriate policies to intervene. Implementing a green fiscal recovery plan can increase economic growth while reducing greenhouse gas emissions. At this time, the environmental audit will appear more important in management, supervision, and evaluation. In addition, the combination of clustering maps, social capital (#1), diffusion of innovation environmental management (#4), and audit committee (#5), is currently the focus of continuous research, and its research trends are likely to continue. The audit committee lasted from 2014 to 2020. This is one of the longest lasting clusters, indicating that it is always the focus of research. Establishing a professional environmental audit department, formulating effective and authoritative audit standards, and cultivating more professional environmental audit talents are also things we must always carry out.

## 5. Conclusions

This study aimed to perform a scientometric study on climate change and environmental audit with the utilization of CiteSpace and 84 document records to obtain some knowledge about the current status of this field and detect its hot topics and research trends. Figure 11 shows the article structure framework.

First, from the perspective of research output, since 2013, the number of papers published in the fields of climate change and environmental audit has been on the rise. From 2013 to 2016, the rate of growth was relatively slow and the number of publications per year remained below 20. Second, based on the research direction, environmental sciences ecology is the most researched direction, followed by business economics, energy fuels, engineering, and meteorology atmospheric sciences. Third, from the perspective of the distribution of journals, the distribution of journals in this field is relatively scattered, and the journal with the most articles published only four articles in this field, accounting for 5% of the total.

Then, through the CiteSpace visualization software, we detect the hot topics and intrinsic relations from the perspective of countries, cited authors, and cited journals. Especially from the analysis of the co-citation authors, we divided the research content of these scholars into six categories by the clustering method and display them in a timeline view. The most important hot spots in the field over last the 9 years are carbon emissions, social capital, energy audit, corporate governance, diffusion of innovation environmental management system, and audit committee. These are the key research contents of climate change and environmental audit. Finally, in order to clarify the research frontiers and trends, we summarize the keywords into three topics by using the keyword co-occurrence network. The keywords in the time zone view clearly reflect the research focus of climate change and environmental audit at each time point. Eventually, combining keyword analysis and cluster analysis, step by step, reveals the key content of environmental audit development. The high-frequency keywords city, policy, dynamics, information, biodiversity, conservation and clustering social capital (#1), diffusion of innovation environmental management (#4), and audit committee (#5) are likely to be the focus of future research. In addition, cities, policies, and adaptability are closely linked to public health. Environmental audit plays a vital role in responding to these measures. Therefore, this article provides an overall structure of research in climate change and environmental audit through scientometrics and information visualization methods. It is committed to future environmental audit that can be more efficient and targeted on the most climate-affected aspects. It is hoped that the majority of scholars can pay attention to the research in this area, and through auditing the environment, put forward more targeted measures to improve the climate.

This study still has some space for improvement. First, this it is limited to the Web of Science database of English journal papers. The non-English journal articles could add another layer of insights to this paper. In addition, the search range of keywords can be further expanded, and the environmental audit can be specifically analyzed in a certain field (such as natural resource audit) for relevant literature analysis, so that its impact on climate change can be analyzed from a specific and microscopic perspective.

## Figures and Tables

**Figure 1 ijerph-19-04142-f001:**
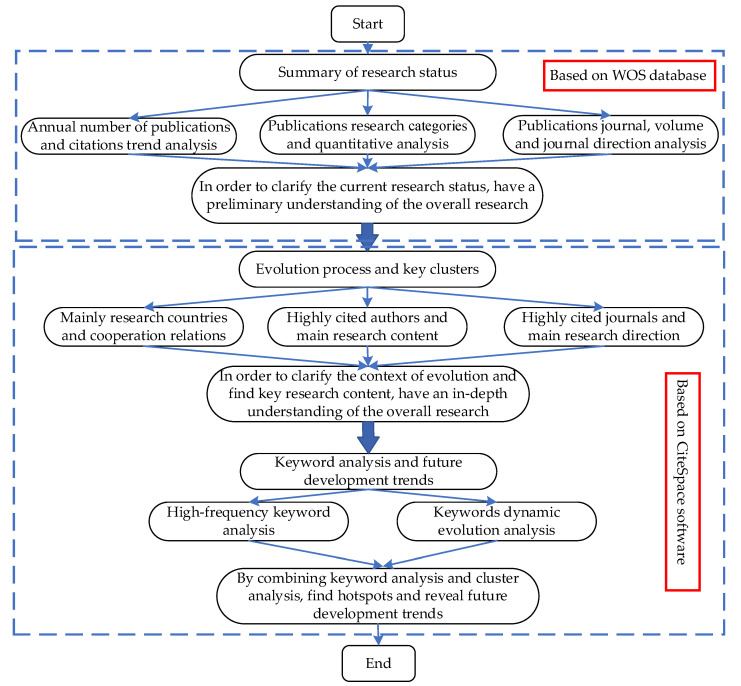
The schematic diagram of the research methods and processes.

**Figure 2 ijerph-19-04142-f002:**
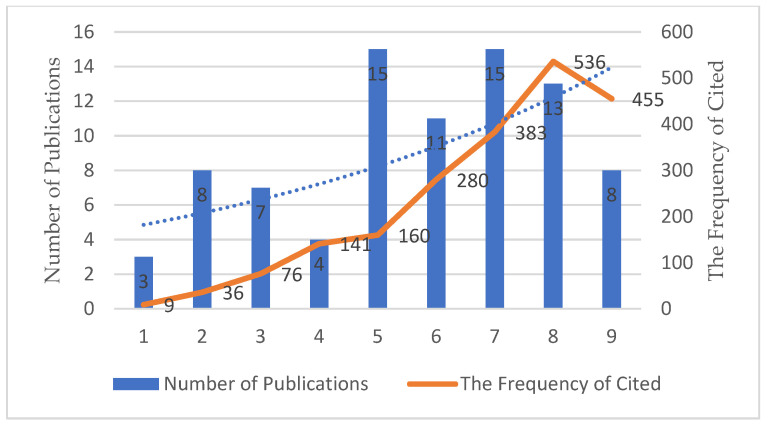
The number of publications and the frequency of cited by year.

**Figure 3 ijerph-19-04142-f003:**
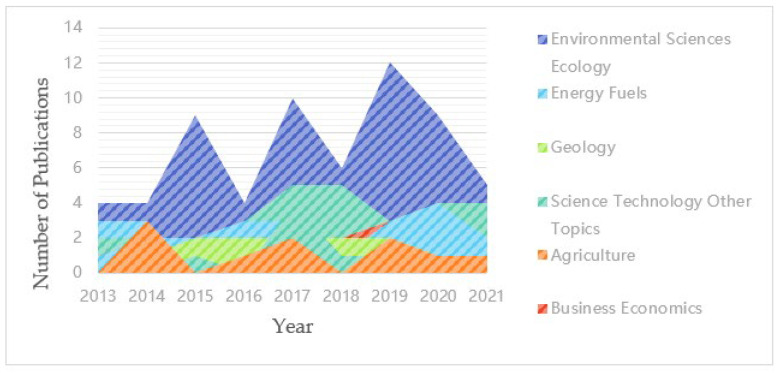
Research categories.

**Figure 4 ijerph-19-04142-f004:**
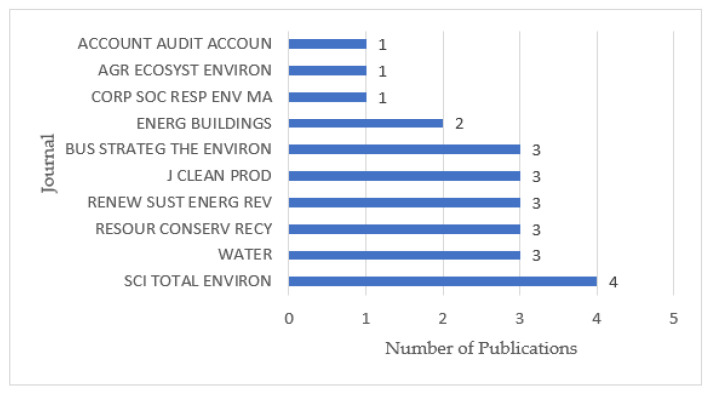
Journal distribution.

**Figure 5 ijerph-19-04142-f005:**
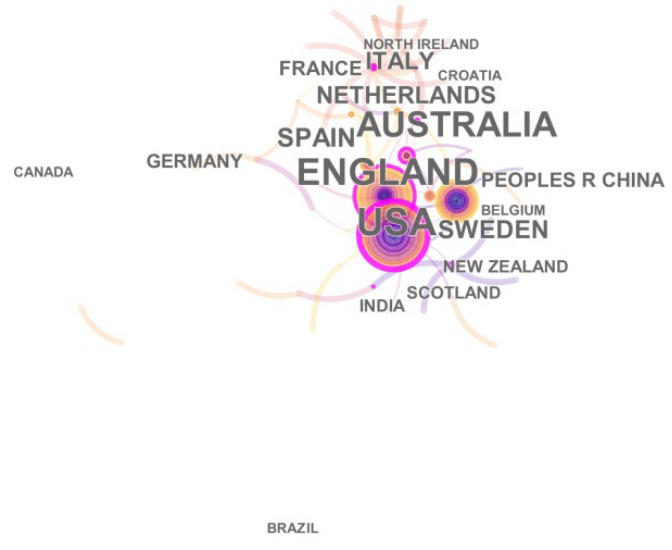
A visualization of the country collaboration network.

**Figure 6 ijerph-19-04142-f006:**
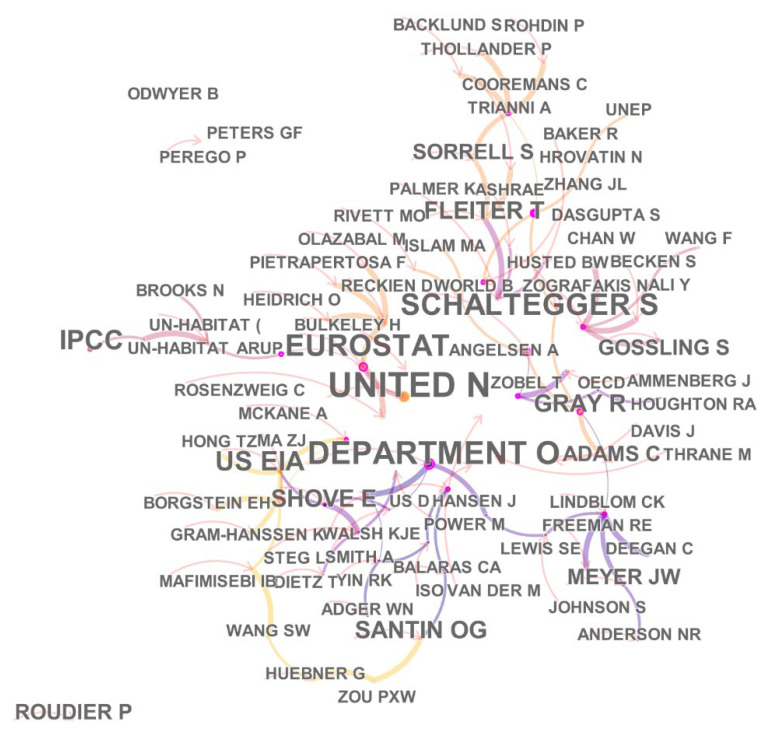
A visualization of the author co-citation network.

**Figure 7 ijerph-19-04142-f007:**
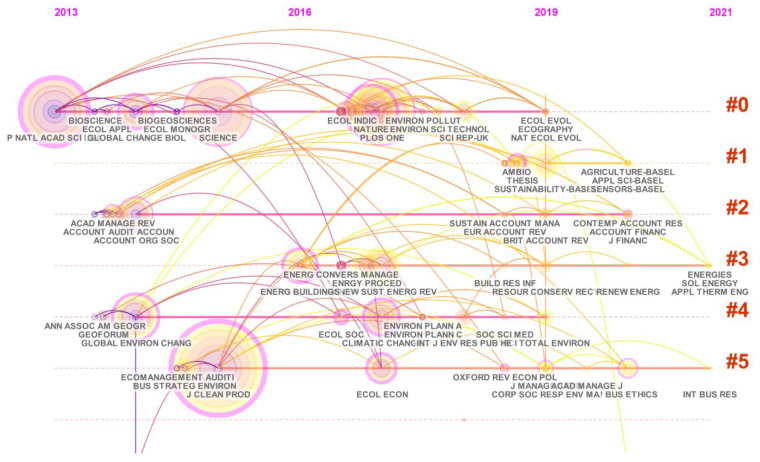
A visualization of the keyword clustering timeline network.

**Figure 8 ijerph-19-04142-f008:**
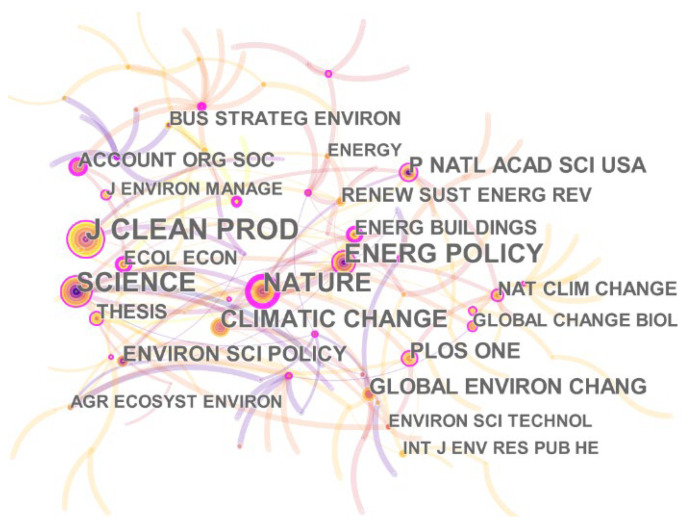
A visualization of the journal co-citation network.

**Figure 9 ijerph-19-04142-f009:**
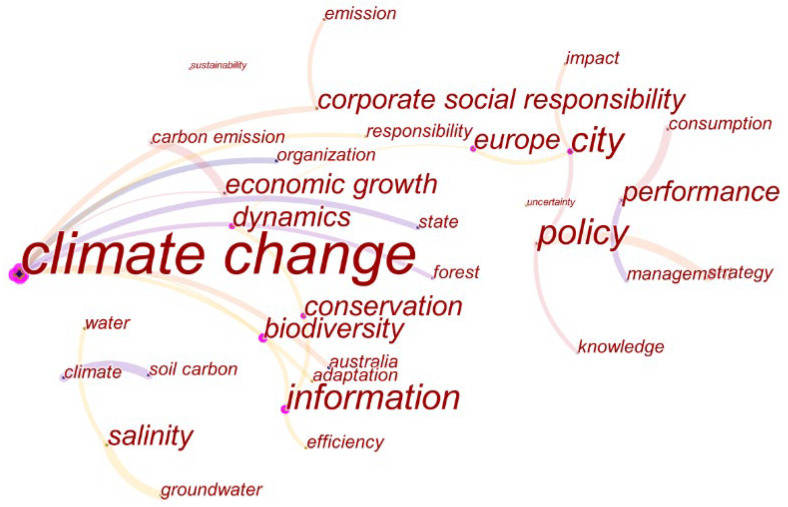
A visualization of the keyword co-occurrence network.

**Figure 10 ijerph-19-04142-f010:**
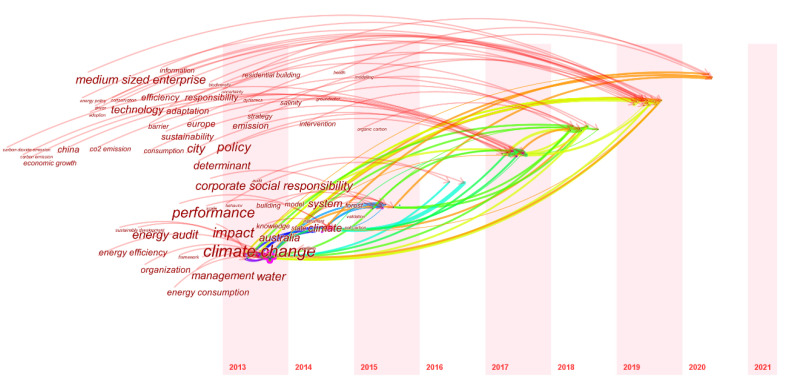
A visualization of the keyword time zone map.

**Figure 11 ijerph-19-04142-f011:**
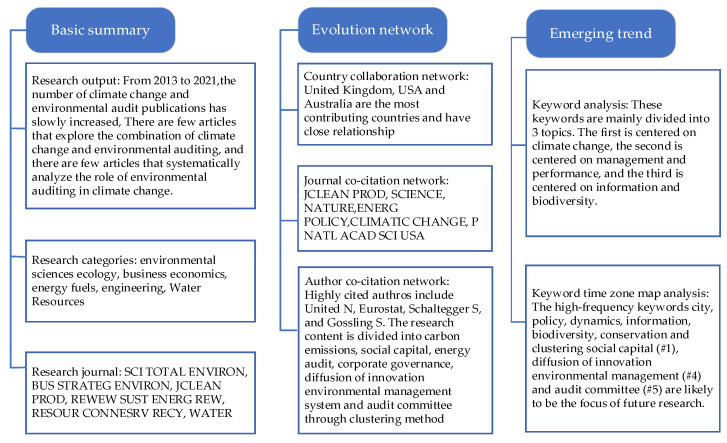
Article structure framework.

**Table 1 ijerph-19-04142-t001:** Keywords and their detailed information.

Frequency	Centrality	Keywords	Frequency	Centrality	Keywords
38	0.62	Climate Change	2	0.46	Biodiversity
6	0.24	Impact	2	0.42	Conservation
6	0.31	City	2	0.18	Carbon Emission
4	0.17	Policy	2	0.02	Strategy
4	0.11	Performance	2	0.00	Salinity
4	0.00	Management	2	0.00	Soil Carbon
4	0.39	Dynamics	2	0.00	Responsibility
4	0.12	Adaptation	2	0.05	Knowledge
4	0.00	Uncertainty	2	0.09	Economic Growth
3	0.49	Information	2	0.00	Supply Chain Management

**Table 2 ijerph-19-04142-t002:** Countries and their detailed information.

Frequency	Centrality	Countries
23	0.62	UK
17	0.21	Australia
14	0.21	USA
5	0.08	Spain
4	0.06	Netherlands
2	0.00	Sweden
2	0.00	Italy
2	0.00	India

**Table 3 ijerph-19-04142-t003:** Top 10 most-cited authors with co-citation frequency.

Frequency	Centrality	Author	Frequency	Centrality	Author
6	0.05	United N.	2	0.36	Zobel T.
4	0.12	Eurostat	2	0.46	Zografakis N.
4	0.28	Schaltegger S.	2	0.01	Meyer J.W.
3	0.06	Gossling S.	2	0.08	Dasgupta S.
3	0	Adams C.	2	0.05	Husted B.W.

**Table 4 ijerph-19-04142-t004:** Summary of the clusters.

Cluster	Periods	Label (LLR)
#0	2013–2019	Carbon Emissions
#1	2018–2020	Social Capital
#2	2016–2021	Energy Audit
#3	2014–2021	Corporate Governance
#4	2013–2019	Diffusion of Innovation Environmental Management System
#5	2014–2020	Audit Committee

**Table 5 ijerph-19-04142-t005:** Top 10 most-cited journals.

Frequency	Centrality	Cited Journals
26	0.12	Journal of Cleaner Production
22	0.19	Science
18	0.50	Nature
18	0.15	Energy Policy
14	0.03	Climatic Change
12	0.17	Proceedings of the National Academy of Sciences of the United States of America
11	0.12	Plos One
11	0.05	Global Environmental Chang-Human and Policy Dimensions
10	0.10	Environmental Science& Policy
9	0.24	Accounting Organizations and Society

**Table 6 ijerph-19-04142-t006:** Summary of the keywords.

Count	Year	Keyword	Count	Year	Keyword
38	2013	Climate Change	4	2019	Dynamics
6	2016	Impact	4	2017	Adaptation
6	2017	City	4	2018	Uncertainty
4	2017	Policy	3	2014	Information
4	2014	Performance	2	2019	Biodiversity
4	2014	Management	2	2019	Conservation

## Data Availability

Not applicable.

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
