# Peer review of "Exploring Knowledge Domain and Emerging Trends in Climate Change and Environmental Audit: A Scientometric Review"

_ijerph, 2022, doi:10.3390/ijerph19074142_

Round 1

Reviewer 1 Report

Dear Authors,

Thank you for you interesting study. Few minor thing should be adjusted:

  • Figure 2 number of years should be uniform with the Figure 3
  • Link between COVID (which is discussed quite a lot) and the study is unclear. I suggest to eliminate this discussion completely and discuss the importance and variety of different environmental audits more widely.
  • Sentence structures and text in general should be adjusted accordingly after this.

Author Response

Response to Reviewer 1 Comments

Paper ID:ijerph-1585730

Dear Editors and Reviewers:

First of all, we sincerely thank you for your time and constructive comments on our manuscript entitled “Exploring Knowledge domain and emerging trends in Climate Change and Environmental Audit:A scientometric review based on CiteSpace analysis” (ID: ijerph-1585730). Those comments are all valuable and very helpful for revising and improving our paper, as well as the important guiding significance to our researches. We have studied these comments carefully and have made correction which we hope meet with approval. Revised portion are marked in red in the paper. The main corrections in the paper and the point-to-point responses to the editor’s comments are as following:

Reviewer 1

Comments and Suggestions for Authors

Thank you for you interesting study. Few minor thing should be adjusted:

Point 1: Figure 2 number of years should be uniform with the Figure 3

Response 1: We have made correction according to the Reviewer’s comments. We revised the number of years in Figure 2 to be consistent with Figure 3. As shown in the figure below. Please check the revised version using the "Track Changes" function in Microsoft Word.

Figure 2. The number of publications and the frequency of cited by year.

Figure 3. Research categories.

Point 2: Link between COVID (which is discussed quite a lot) and the study is unclear. I suggest to eliminate this discussion completely and discuss the importance and variety of different environmental audits more widely.

Response 2: Thank you for your suggestion. We we have removed the part of the article that mentioned the weak link between Covid-19 in this study. In addition, we have added references [2]JiangJJ.The Impact of COVID-19 on Climate-Related Financial Risks-Also on Macroeconomic Policy Response. Wuhan Finance, 2021(01),14-19. to support this point of view on the relationship between COVID and climate change mentioned on line 55. Please check the revised version using the "Track Changes" function in Microsoft Word. .

Point 3: Sentence structures and text in general should be adjusted accordingly after this.

Response 3: We think this is an excellent suggestion. We have checked the full text and adjusted sentence structure and text content. Please check the revised version using the "Track Changes" function in Microsoft Word.

Thank you again for your constructive comments that help us a lot to improve the paper.

Reviewer 2 Report

I advise the authors to consider addressing the following comments:

  1. The name of the program with which the analysis of the information was carried out should not be in the title and in the Abstract because in this way it looks as if the specific program is promoted. Alternatively, I recommend using this title: Exploring Knowledge domain and emerging trends in Climate Change and Environmental Audit: A scientometric review.
  2. Words included in the title should not be iterated as keywords.
  3. In a considerable part of the paper, the authors provide significant information without however citing the literature sources from which this information was derived. The tendency not to use citations is very intense in the first two paragraphs of the Introduction and in the Discussion. Hence, the authors need to ensure that appropriate literature sources are used.
  4. The abstract should not exceed 200 words in line with the author instructions of Int. J. Environ. Res. Public Health.
  5. The authors need to justify why their analysis of academic publications focused explicitly on the period between 2013 and 2021.
  6. In lines 150-151, the authors state that “Articles analyzing the impact of environmental audit on climate change have only appeared in recent years”. However, this does not hold true because articles on this topic had been published in previous years. What has really happened is that in recent years the publication of such articles has increased significantly. The authors should thus revise this statement accordingly.
  7. Sentences in lines 53-56 convey important information which should be supported and referenced with the appropriate references.
  8. Lines 107-116 should be removed from the Introduction and be transferred to the Methodology section.
  9. In terms of the English language, the paper is overall well-written, however, minor editing should be carried out since there are a few minor errors.

Author Response

Response to Reviewer 2 Comments

Paper ID:ijerph-1585730

Dear Editors and Reviewers:

First of all, we sincerely thank you for your time and constructive comments on our manuscript entitled “Exploring Knowledge domain and emerging trends in Climate Change and Environmental Audit:A scientometric review based on CiteSpace analysis” (ID: ijerph-1585730). Those comments are all valuable and very helpful for revising and improving our paper, as well as the important guiding significance to our researches. We have studied these comments carefully and have made correction which we hope meet with approval. Revised portion are marked in red in the paper. The main corrections in the paper and the point-to-point responses to the editor’s comments are as following:

Reviewer 2

Comments and Suggestions for Authors

I advise the authors to consider addressing the following comments:

Point 1: The name of the program with which the analysis of the information was carried out should not be in the title and in the Abstract because in this way it looks as if the specific program is promoted. Alternatively, I recommend using this title: Exploring Knowledge domain and emerging trends in Climate Change and Environmental Audit: A scientometric review.

Response 1: Expert opinions are very meaningful. In response to this issue, according to the opinions of experts, the title of the article is revised to:Exploring Knowledge domain and emerging trends in Climate Change and Environmental Audit: A scientometric review. Please check the revised version using the "Track Changes" function in Microsoft Word.

Point 2: Words included in the title should not be iterated as keywords.

Response 2: The expert's opinion is very correct, we have revised the keyword part of the article according to this suggestion. Please check the revised version using the "Track Changes" function in Microsoft Word.

Point 3: In a considerable part of the paper, the authors provide significant information without however citing the literature sources from which this information was derived. The tendency not to use citations is very intense in the first two paragraphs of the Introduction and in the Discussion. Hence, the authors need to ensure that appropriate literature sources are used.

Response 3: The expert opinion is very correct, and we have made a source statement for the key parts mentioned in the paper. In particular, the corresponding references are added in the introduction section, and the source of the literature is given. In addition, the corresponding references and literature sources are also supplemented in the Discussion section. They are documents [1-6], [10-12]. Please check the revised version using the "Track Changes" function in Microsoft Word.

[1]Indhumathi K, Kumar K S. A review on prediction of seasonal diseases based on climate change using big data[J]. Materials Today: Proceedings, 2021, 37: 2648-2652.

[2]JiangJJ.The Impact of COVID-19 on Climate-Related Financial Risks-Also on Macroeconomic Policy Response. Wuhan Finance, 2021(01),14-19.

[3]IPCC, 'AR5 Synthesis Report: Climate Change 2014', < https://www.ipcc.ch/report/ar5/syr/>

[4]IPCC, 'SPECIAL REPORT:Global Warming of 1.5 ºC', < https://www.ipcc.ch/sr15/>

[5]Watson M , Emery A . Environmental management and auditing systems[J]. Managerial Auditing Journal, 2004, 19(7):916-928.

[6]YangZH. Discussion on the Definition of Environmental Performance Audit. Accounting Newsletter, 2009(10):25-26.

[10]Wider M , Szymon Szewrański, Kazak J K . Environmental Carrying Capacity Assessment—the Policy Instrument and Tool for Sustainable Spatial Management[J]. Frontiers in Environmental Science, 2020, 8:579838.

[11]Widodo B ,  Lupyanto R , Sulistiono B , et al. Analysis of Environmental Carrying Capacity for the Development of Sustainable Settlement in Yogyakarta Urban Area[J]. Procedia Environmental Sciences, 2015, 28:519-527.

[12]Li Y , Ye H , Sun X , et al. Coupling Analysis of the Thermal Landscape and Environmental Carrying Capacity of Urban Expansion in Beijing (China) over the Past 35 Years[J]. Sustainability, 2021, 13(2):584.

Point 4: The abstract should not exceed 200 words in line with the author instructions of Int. J. Environ. Res. Public Health.

Response 4: Thank you for your opinion, we have revised the abstract part of the article Please check the revised version using the "Track Changes" function in Microsoft Word.

Point 5: The authors need to justify why their analysis of academic publications focused explicitly on the period between 2013 and 2021.

Response 5: Thank you for your suggestion. We explain why the years analyzed are concentrated between 2013 and 2021, in line 170 of the article. Please check the revised version using the "Track Changes" function in Microsoft Word.

Point 6: In lines 150-151, the authors state that “Articles analyzing the impact of environmental audit on climate change have only appeared in recent years”. However, this does not hold true because articles on this topic had been published in previous years. What has really happened is that in recent years the publication of such articles has increased significantly. The authors should thus revise this statement accordingly.

Response 6: The expert opinion is very correct. As suggested by the reviewer, we revise accordingly that "Articles analyzing the impact of environmental audit on climate change have only appeared in recent years" has been revised accordingly". Please check the revised version using the "Track Changes" function in Microsoft Word.

Point 7: Sentences in lines 53-56 convey important information which should be supported and referenced with the appropriate references.

Response 7: The expert opinion is very correct, We have added corresponding references according to the opinions of the experts, and have added the following papers into the References Section of my paper and referred to the paper “[1]Indhumathi K, Kumar K S. A review on prediction of seasonal diseases based on climate change using big data[J]. Materials Today: Proceedings, 2021, 37: 2648-2652. [2]JiangJJ.The Impact of COVID-19 on Climate-Related Financial Risks-Also on Macroeconomic Policy Response. Wuhan Finance, 2021(01),14-19.” Please check the revised version using the "Track Changes" function in Microsoft Word.

Point 8: Lines 107-116 should be removed from the Introduction and be transferred to the Methodology section.

Response 8: We think this is an excellent suggestion. We have removed lines 107-116 from the introduction and added a Methods section. Please check the revised version using the "Track Changes" function in Microsoft Word.

Point 9: In terms of the English language, the paper is overall well-written, however, minor editing should be carried out since there are a few minor errors.

Response 9: Thank you for your suggestion. We have carefully checked the manuscript and corrected the errors accordingly. Please check the revised version using the "Track Changes" function in Microsoft Word.

Thank you again for your constructive comments that help us a lot to improve the paper.

Reviewer 3 Report

The paper presents an interesting research which fits to the scope of the journal, however, I have some comments which should be taken into consideration before publishing.

  1. Abstract should be shortened according to the requirements of MDPI format - about 200 words.
  2. “There is no denying that the coronavirus disease (COVID-19) pandemic is related to the climate crisis.” Is it an opinion of authors or scientifically proved fact?
  3. There are paragraphs of the introduction which are not supported by references. In the first paragraph IPCC report is mentioned but not listed in references. In the second paragraph information about environmental audit promotion in 1960s, but not proven by any source. Such kind of information needs references.
  4. Very limited literature review resulted in poor presentation of state of art and defining gap of knowledge. One of the possible options of expanding literature review could be referring to findings from incorporating worldwide known metrics in environmental management, like ecological footprint and environmental carrying capacity (see for instance: Environmental Carrying Capacity Assessment—the Policy Instrument and Tool for Sustainable Spatial Management. Frontiers in Environmental Science, 8:579838.; Analysis of Environmental Carrying Capacity for the Development of Sustainable Settlement in Yogyakarta Urban Area. Procedia Environmental Sciences, 28: 519-527; Coupling Analysis of the Thermal Landscape and Environmental Carrying Capacity of Urban Expansion in Beijing (China) over the Past 35 Years. Sustainability, 13(2): 584; and other papers in that field).
  5. Resolution of Figure 1 should be improved.
  6. Discussion section focuses on analyzing obtained results, however, comparative discussion with confronting them with other studies is very limited.

Author Response

Response to Reviewer 3 Comments

Paper ID:ijerph-1585730

Dear Editors and Reviewers:

First of all, we sincerely thank you for your time and constructive comments on our manuscript entitled “Exploring Knowledge domain and emerging trends in Climate Change and Environmental Audit:A scientometric review based on CiteSpace analysis” (ID: ijerph-1585730). Those comments are all valuable and very helpful for revising and improving our paper, as well as the important guiding significance to our researches. We have studied these comments carefully and have made correction which we hope meet with approval. Revised portion are marked in red in the paper. The main corrections in the paper and the point-to-point responses to the editor’s comments are as following:

Reviewer 3

Comments and Suggestions for Authors

Point 1: Abstract should be shortened according to the requirements of MDPI format - about 200 words.

Response 1: Thank you for your opinion, we have revised the abstract part of the article Please check the revised version using the "Track Changes" function in Microsoft Word.

Point 2: There is no denying that the coronavirus disease (COVID-19) pandemic is related to the climate crisis.” Is it an opinion of authors or scientifically proved fact?

Response 2: Thank you for your suggestion, the "There is no denying that the coronavirus disease (COVID-19) pandemic is related to the climate crisis." written in the article is a scientific fact, for this problem, and we have cited relevant references.”[2]JiangJJ.The Impact of COVID-19 on Climate-Related Financial Risks-Also on Macroeconomic Policy Response. Wuhan Finance, 2021(01),14-19.” Please check the revised version using the "Track Changes" function in Microsoft Word.

Point 3: There are paragraphs of the introduction which are not supported by references. In the first paragraph IPCC report is mentioned but not listed in references. In the second paragraph information about environmental audit promotion in 1960s, but not proven by any source. Such kind of information needs references.

Response 3: As suggested by the expert, we have added more references to support this idea. We have added the following references for support. “[3]IPCC, 'AR5 Synthesis Report: Climate Change 2014', < https://www.ipcc.ch/report/ar5/syr/>;[4]IPCC, 'SPECIAL REPORT:Global Warming of 1.5 ºC', < https://www.ipcc.ch/sr15/>;[6]YangZH. Discussion on the Definition of Environmental Performance Audit. Accounting Newsletter, 2009(10):25-26.”Please check the revised version using the "Track Changes" function in Microsoft Word.

Point 4: Very limited literature review resulted in poor presentation of state of art and defining gap of knowledge. One of the possible options of expanding literature review could be referring to findings from incorporating worldwide known metrics in environmental management, like ecological footprint and environmental carrying capacity (see for instance: Environmental Carrying Capacity Assessment—the Policy Instrument and Tool for Sustainable Spatial Management. Frontiers in Environmental Science, 8:579838.; Analysis of Environmental Carrying Capacity for the Development of Sustainable Settlement in Yogyakarta Urban Area. Procedia Environmental Sciences, 28: 519-527; Coupling Analysis of the Thermal Landscape and Environmental Carrying Capacity of Urban Expansion in Beijing (China) over the Past 35 Years. Sustainability, 13(2): 584; and other papers in that field).

Response 4: We have revised this problem according to the reviewer’s valuable suggestion and have added the following papers into the References Section of my paper and added more references on ecological footprint and environmental carrying capacity into the INTRODUCTION part in the revised manuscript. Corresponding references in the article [10]Wider M , Szymon Szewrański, Kazak J K . Environmental Carrying Capacity Assessment—the Policy Instrument and Tool for Sustainable Spatial Management[J]. Frontiers in Environmental Science, 2020, 8:579838., [11]Widodo B ,  Lupyanto R ,  Sulistiono B , et al. Analysis of Environmental Carrying Capacity for the Development of Sustainable Settlement in Yogyakarta Urban Area[J]. Procedia Environmental Sciences, 2015, 28:519-527., [12]Li Y , Ye H , Sun X , et al. Coupling Analysis of the Thermal Landscape and Environmental Carrying Capacity of Urban Expansion in Beijing (China) over the Past 35 Years[J]. Sustainability, 2021, 13(2):584. Please check the revised version using the "Track Changes" function in Microsoft Word.

Point 5: Resolution of Figure 1 should be improved

Response 5: Thank you for your suggestion. we have adjusted the resolution of the picture according to the opinion. A clearer picture is shown below. Please check the revised version using the "Track Changes" function in Microsoft Word.

Round 2

Reviewer 3 Report

The paper has been corrected according to my previous comments and in my opinion it can be published in the current form.

This manuscript is a resubmission of an earlier submission. The following is a list of the peer review reports and author responses from that submission.